# Evaluating Inductive Parameter-Based Transfer Learning with Deep Neural Networks for Wind Forecasting in Corsica

## Abstract

This study assesses the effectiveness of various transfer learning strategies for wind speed forecasting across meteorological stations in Corsica using deep neural networks. Leveraging inductive parameter-based transfer, models are transferred based on geographic proximity, topographic classification, dominant wind direction, and random assignment. Several architectures are evaluated, including recurrent, convolutional, attention-based, and dense networks. Results indicate that structured transfer strategies do not consistently outperform non-transfer baselines. This lack of improvement can be largely attributed to significant distributional differences in wind speed across stations, which hinder model transferability. These findings highlight the challenges posed by domain shift in a geographically heterogeneous insular context and emphasize the need for more refined similarity criteria, hybrid transfer strategies, and spatially-aware modeling, notably through graph neural networks. The results also call for a critical reassessment of commonly held assumptions about the benefits of transfer learning in complex meteorological environments.

## 1 Introduction

Accurate wind speed forecasting is a critical challenge for various industrial and environmental applications, particularly in the management of renewable energy and the mitigation of climate-related risks Jiang et al. (2021); Khodayar et al. (2017). However, the comprehensive collection of high-resolution meteorological data remains complex, especially in regions such as Corsica. Transfer learning, by reusing models previously trained on similar source domains, offers a promising approach to overcome these constraints while reducing computational costs Wellens et al. (2021). It provides a methodological response to the limitations of conventional supervised learning, particularly in contexts with limited labeled data, feature space divergences, or distribution mismatches between training and testing sets Gholizade et al. (2025). By leveraging a model pre-trained on a source domain, transfer learning enables improved performance on a target domain while reducing data requirements and computational costs Sankari & Kumar (2023).

Several taxonomies have been proposed to structure the literature on transfer learning. From the perspective of the *label space*, three paradigms are typically distinguished: *inductive* transfer, involving labeled data in both domains; *transductive* transfer, where only the source data are labeled; and *unsupervised* transfer, where no labels are available Gholizade et al. (2025). The relationship between these paradigms and the similarity between source and target domains has been emphasized in previous studies Sankari & Kumar (2023). In terms of the feature space, a distinction is made between *homogeneous* settings, where the features are identical but may differ in distribution, and *heterogeneous* settings, where the feature spaces differ Gholizade et al. (2025); Sankari & Kumar (2023).

Transfer mechanisms can also be categorized by the nature of the transferred knowledge. The main approaches include: (i) *instance-based transfer*, which involves selecting or reweighting relevant source examples; (ii) *feature-based transfer*, aiming to project data into a common representative space; (iii) *parameter-based transfer*, which reuses weights from a source model; and (iv) *relation-based transfer*, which exploits structural similarities between domains Gholizade et al. (2025); Al-

Hajj et al. (2023). In all cases, the chosen strategy depends on data availability, task similarity, and the application domain. A poor alignment between source and target can lead to *negative transfer*, resulting in degraded performance Zhang et al. (2021).

In the field of wind speed forecasting, transfer learning has been used to address dataset heterogeneity across sites or tasks. For instance, Qureshi & Khan (2018) introduced an inter-site framework based on sparse autoencoders guided by a deep belief network. This adaptive system (ATL-DNN) dynamically adjusts the transferred representations according to local characteristics. Oh et al. (2022) proposed an approach involving partial layer sharing in a C-LSTM model, demonstrating accuracy improvements for data-scarce sites. In an instance-based transfer context, Cai et al. (2019) proposed a source selection mechanism to mitigate negative transfer effects and improve quantile forecasting via GBDT. Task transfer has also been explored: Qureshi & Khan (2018) studied the transition from power to wind speed prediction, while Chen (2022) employed *knowledge distillation* to transfer representations from a complex teacher model to a lightweight student model.

Regarding *neural architectures*, several families have been adopted in these approaches. Recurrent networks such as LSTM and BiLSTM are employed for their ability to model temporal dependencies Oh et al. (2022); Chen (2022), while autoencoders are used to produce compressed and transferable latent representations Qureshi & Khan (2018); Oh et al. (2022). Additionally, some contributions combine transfer learning with ensemble models such as GBDT Cai et al. (2019) or optimized Adaboost Chen (2022), highlighting the complementarity between statistical robustness and generalization capacity.

Overall, these works converge on a common goal: maximizing predictive performance in low-data scenarios while reducing the computational resources required for model training. It is within this perspective that our contribution is situated, leveraging transfer learning to reduce the size of training datasets needed in meteorological contexts, and thereby decreasing the computational costs associated with predictive model training.

This study presents an analysis of structured transfer learning strategies applied to wind speed prediction across a network of meteorological stations in Corsica. We specifically compare strategies based on distance, topographic classification, directional wind speed dominance, and random transfer against the same neural architecture without transfer learning. Several neural architectures are evaluated, encompassing multiple paradigms: recurrent networks, convolutional-recurrent networks, multi-layer perceptrons (dense), and attention mechanisms.

## 2 DATA

This section presents the dataset used in our study, comprising time series of wind speed measurements collected from multiple meteorological stations, as well as the preprocessing procedures applied to ensure the quality and consistency of the data before their integration into the predictive models. All station data originate from official Météo France records available at : meteo.data.gouv.fr

### 2.1 STATION DISTRIBUTION

Our study is based on the analysis of meteorological stations located throughout the island of Corsica. These stations are represented as red dots in Figure 1 left part.

Out of the 98 existing stations, we selected 22 based on two main criteria: the availability of wind speed measurements and the length of their historical time series. Specifically, we retained stations with more than 80,000 hours of data (approximately 9 years), ensuring significant temporal continuity and minimizing the need for interpolation. This threshold was determined through a comparative analysis of all available stations: a distinct group stood out with dense and regular temporal coverage, whereas others exhibited frequent and substantial gaps in their records.

The right part of figure 1 illustrates Corsica's topography using a color-coded altitude map. This visualization highlights the island's geographical diversity, notably the alternation between coastal areas and mountainous regions, which are major factors influencing wind dynamics Grante et al. (2025). The location of the meteorological stations was analyzed in relation to this topography to ensure good representativeness across different geographical contexts.

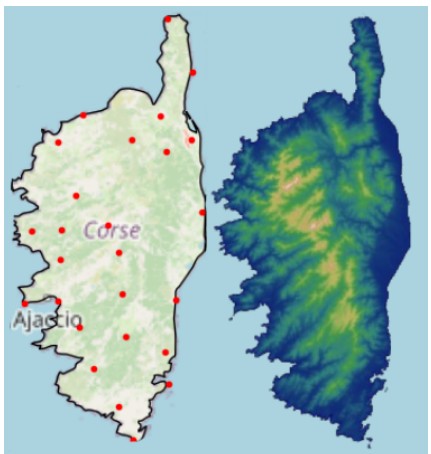

Figure 1: Weather Stations Distribution and Topography of Corsica

## 2.2 FEATURES

The meteorological variables used in our study are listed in Table 1. These constitute the main input features for our predictive models and were selected in line with prior work such as Ryu et al. (2022); Yang et al. (2023); Tang et al. (2020); Geng et al. (2020).

Table 1: Features of the meteorological station dataset

| Name | Units | Description |
|---|---|---|
| FF | $m.s^{-1}$ | Wind speed |
| $DD_{cos}$ | none | Cosine of wind direction (FF) |
| $DD_{sin}$ | none | Sine of wind direction (FF) |
| FXI | $m.s^{-1}$ | Maximum wind speed |
| U | % | Relative humidity |
| T | $°C$ | Temperature at 2 meters |

We additionally included four features to decompose the day and hour of measurement, enabling non-temporally-optimized models to better capture temporal dynamics Baile & Muzy (2022). Furthermore, two features were added to better represent wind direction. Since the wind direction angle *DD* ranges from 0 to 360°, values like 1° and 359° are close in physical meaning but numerically distant. We chose to encode this circular characteristic directly in the input features.

For $S$, the vector representing temporal features, we define:

$$S = [S_1, S_2, S_3, S_4] = \left[\cos\left(2\pi \frac{h}{24}\right), \ \sin\left(2\pi \frac{h}{24}\right), \ \cos\left(2\pi \frac{d}{365.25}\right), \ \sin\left(2\pi \frac{d}{365.25}\right)\right] \quad (1)$$

where $h$ is the hour of the day and $d$ is the day of the year.

For the wind direction DD, we compute:

$$\theta = \pi \frac{\text{DD}}{180} \quad (2)$$

then

$$\text{DD}_{\cos} = \cos(\theta), \quad \text{DD}_{\sin} = \sin(\theta) \quad (3)$$

The final input dataset is composed as follows:

$$\text{X} = [\text{FF}, \ \text{DD}_{\cos}, \ \text{DD}_{\sin}, \ \text{FXI}, \ U, \ T, \ S_1, \ S_2, \ S_3, \ S_4] \quad (4)$$

Each station has its own dataset, denoted $X_{station}$. For each $X_{station}$, missing values were linearly interpolated. Subsequently, we standardized *FF*, *FXI*, *U*, and *T* to align their scales and to improve

training efficiency and stability by reducing overfitting. Variables from the vector $S$ as well as $DD_{\sin}$ and $DD_{\cos}$ are already bounded in the range $[-1, 1]$ and thus were not normalized or standardized.

## 3 METHODOLOGY

This section presents the deep learning models used for wind speed forecasting, as well as the various transfer learning strategies employed.

### 3.1 METRICS

The root mean square error (RMSE) was selected as the evaluation metric for this study. It is generally defined as:

$$\text{RMSE} = \sqrt{\frac{1}{n} \sum_{i=1}^{n} (y_i - \hat{y}_i)^2} \tag{5}$$

It represents the average Euclidean distance between the true value $y$ and the model prediction $\hat{y}$. This metric penalizes larger errors more heavily, making it sensitive to outliers. A low RMSE indicates that the model predictions are close to the true values, while a high RMSE reflects significant discrepancies between predictions and observations.

In order to evaluate the effectiveness of the transfer learning strategies, we compare them to an identical architecture trained without transfer over the same historical period. The gain is defined as the relative reduction in the RMSE, expressed as a percentage:

$$\text{Gain} (\%) = \frac{\text{RMSE}_{\text{reference}} - \text{RMSE}_{\text{transfer}}}{\text{RMSE}_{\text{reference}}} \times 100 \tag{6}$$

### 3.2 MODELS

This subsection introduces the neural architectures selected for comparative analysis. Figure 2 provides a visual representation of the models implemented in this study. Our selection spans a broad range of neural network paradigms, including a Convolutional-LSTM model combining convolutional and recurrent networks, a Convolutional-Dense model integrating convolutional layers with fully connected layers, encoder architectures using the attention mechanism from the Transformer encoder Vaswani et al. (2017), standard LSTM architectures representing recurrent networks Hochreiter & Schmidhuber (1997), and feed-forward neural networks (FFN).

In our approach, all convolutional neural network architectures process the meteorological time series through one-dimensional (1D) convolutions, a choice dictated by the inherently sequential nature of the data. This technical decision aligns with the temporal structure of the variables, where local correlations manifest along the time axis. To ensure a fair comparison while preserving the structural integrity of each model, architectural characteristics (network depth, neuron density, dimensionality of hidden layers) were individually calibrated. This differentiated approach avoids enforcing parametric uniformity across models.

### 3.3 TRANSFER LEARNING

For this comparative study, we focused on inductive transfer learning strategies using parameter transfer by reusing and adjusting the weights of models previously trained (fine-tuning). Since each station has its own forecasting model, we transferred knowledge from one station to another. We compared four geographically informed transfer strategies:

1. Random transfer
2. Topographic classification-based transfer
3. Distance-based transfer
4. Dominant wind direction-based transfer

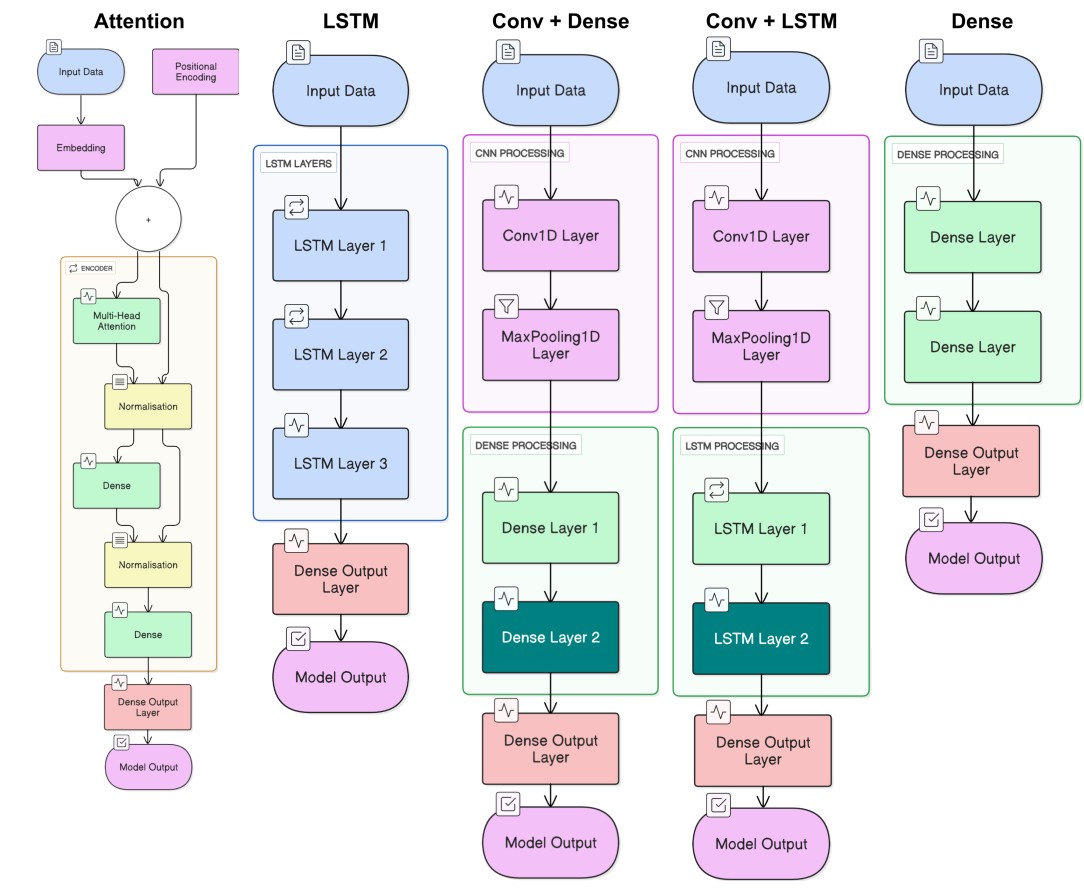

Figure 2: Architecture of the models used

Random transfer is used as a second baseline for evaluating the effectiveness of the transfer strategies. If a transfer strategy fails to outperform random transfer in terms of RMSE, its relevance and ability to leverage structural similarities between stations is questionable. Conversely, a significantly lower RMSE compared to random transfer supports the quality of the considered strategy.

For the topographic classification-based transfer, we initially distinguished three terrain types: mountain, plain, and coastal. However, a joint analysis of the map and the island's topography led us to merge the plain and coastal categories due to their strong topographic correlation. This resulted in a binary classification between mountain and plain-coast zones, using an altitude threshold of 300 meters. The central panel of Figure 3 illustrates this classification, where stations below the threshold are shown in blue (plain-coast), and those above in red (mountain).

For the distance-based transfer, knowledge is transferred to minimize the distance between source and target stations, as shown on the left panel of Figure 3. For each target station, we select the three nearest source stations using the Haversine distance between two geographical points $(\phi_1, \lambda_1)$ and $(\phi_2, \lambda_2)$, defined as:

$$d = 2R \cdot \arcsin\left(\sqrt{\sin^2\left(\frac{\phi_2 - \phi_1}{2}\right) + \cos(\phi_1) \cdot \cos(\phi_2) \cdot \sin^2\left(\frac{\lambda_2 - \lambda_1}{2}\right)}\right) \qquad (7)$$

where:

- $\phi_1, \phi_2$ are the latitudes in radians,
- $\lambda_1, \lambda_2$ are the longitudes in radians,
- $R$ is the Earth's radius,

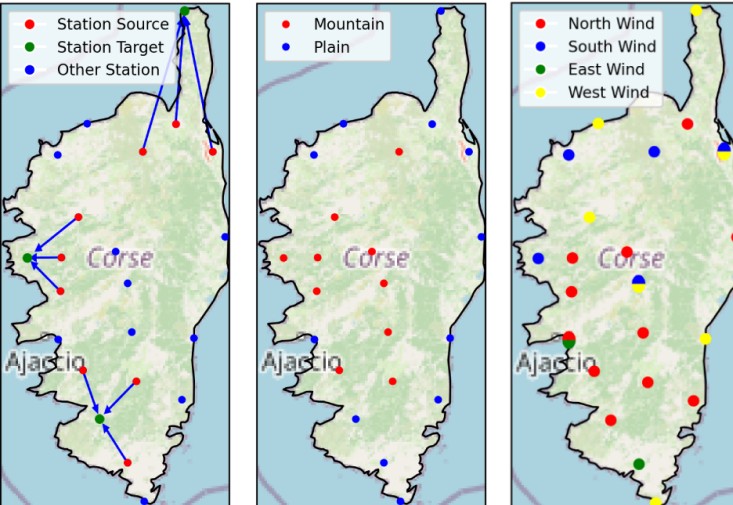

Figure 3: Map of meteorological stations. Left: visualization of transfers between stations. Target stations are shown in green, source stations in red, and other stations in blue. Blue arrows represent transfers from source to target stations. Center: topographic classification of stations. Mountain stations are shown in red; plain/coastal stations in blue. Right: station classification according to dominant wind directions in the study area. Stations are color-coded according to their dominant wind direction: red for north, blue for south, green for east, and yellow for west. Bi-colored stations correspond to cases where the prevailing wind lies between two cardinal directions (e.g., northeast), and are therefore classified into both associated directions.

- $d$ is the distance between the two points.

For the dominant wind direction-based classification, stations were grouped according to the most frequent wind direction over the full observation period. When a station exhibited two dominant directions of comparable intensity, it was assigned to both groups. The right panel of figure 3 illustrates the spatial distribution of stations according to this classification. Bicolored points represent stations associated with two dominant directions. A station with a wind rose showing a shared maximum between north and northeast is considered as exposed to a northerly wind only. The same applies to other intercardinal directions.

Let $E$ denote the set of all stations. Each strategy defines a subset $E' \subseteq E$ on which the transfer is applied. The transfer occurs in a random order among stations in $E'$: a source station $A$ can transfer its knowledge to a target station $B$ if and only if $A, B \in E'$. The source model is a neural network with the same architecture, trained on the full training dataset of station $A$.

### 3.4 EXPERIMENTATION

The models are trained using the Adam optimizer with an initial learning rate of $10^{-3}$. During fine-tuning — the adaptation phase of a model pre-trained on another station — the learning rate is reduced to $10^{-4}$ to promote more stable convergence. In practice, fine-tuning was conducted year by year, progressively extending the historical window: starting from 2020–2020, then 2019–2020, 2018–2020, and so forth, up to 2015–2020. For clarity, we report only the two extreme cases (2020–2020 and 2015–2020). When computing the gain of transfer learning defined in equation 6 with respect to the reference model, the baseline is always trained on the same historical window as the fine-tuned model. Training runs for up to 50 epochs, with early stopping based on validation loss and a patience of 5 epochs.

Performance is evaluated on an independent test set covering the period from 01/01/2023 to 31/12/2023. The validation set spans the year 2022. The training set varies depending on the target year and may extend from 01/01/2015 to 31/12/2021, for a maximum of seven years. Models operate with a sliding input window based on a history of three time steps.

Table 2: Summary of average performance per architecture / strategy pair from (h+1 to h+24): mean of RMSE and gain in % with fine-tuning performed using either 2015 (data from 2015–2020) or 2020 (data from 2020–2020) as historical year.

| Model | Strategy | RMSE 2015 | RMSE 2020 | Gain 2015 (%) | Gain 2020 (%) |
|---|---|---|---|---|---|
| attention | distance | 0.8423 | 0.8561 | -0.33 | -0.63 |
| attention | east | 0.8702 | 0.8835 | -3.65 | -3.85 |
| attention | mountain | 0.8353 | 0.8532 | 0.50 | -0.30 |
| attention | north | 0.8207 | 0.8343 | 2.25 | 1.92 |
| attention | west | 0.8447 | 0.8566 | -0.62 | -0.69 |
| attention | plain | 0.8371 | 0.8516 | 0.29 | -0.11 |
| attention | random | 0.8373 | 0.8558 | 0.27 | -0.60 |
| attention | south | 0.8604 | 0.8688 | -2.49 | -2.13 |
| conv_dense | distance | 0.8448 | 0.8579 | -0.47 | -0.47 |
| conv_dense | east | 0.8731 | 0.8859 | -3.83 | -3.74 |
| conv_dense | mountain | 0.8427 | 0.8559 | -0.22 | -0.23 |
| conv_dense | north | 0.8243 | 0.8417 | 1.96 | 1.42 |
| conv_dense | west | 0.8482 | 0.8608 | -0.87 | -0.81 |
| conv_dense | plain | 0.8414 | 0.8563 | -0.07 | -0.28 |
| conv_dense | random | 0.8404 | 0.8575 | 0.05 | -0.42 |
| conv_dense | south | 0.8642 | 0.8740 | -2.78 | -2.36 |
| dense | distance | 0.8489 | 0.8654 | -1.03 | -1.61 |
| dense | east | 0.8688 | 0.8804 | -3.39 | -3.38 |
| dense | mountain | 0.8396 | 0.8552 | 0.08 | -0.42 |
| dense | north | 0.8239 | 0.8441 | 1.95 | 0.89 |
| dense | west | 0.8480 | 0.8569 | -0.91 | -0.62 |
| dense | plain | 0.8393 | 0.8587 | 0.12 | -0.84 |
| dense | random | 0.8404 | 0.8637 | -0.01 | -1.42 |
| dense | south | 0.8609 | 0.8753 | -2.45 | -2.78 |
| conv_lstm | distance | 0.8466 | 0.8620 | -0.90 | -0.99 |
| conv_lstm | east | 0.8737 | 0.8864 | -4.12 | -3.86 |
| conv_lstm | mountain | 0.8405 | 0.8550 | -0.17 | -0.17 |
| conv_lstm | north | 0.8232 | 0.8403 | 1.90 | 1.55 |
| conv_lstm | west | 0.8474 | 0.8596 | -0.99 | -0.72 |
| conv_lstm | plain | 0.8417 | 0.8558 | -0.31 | -0.27 |
| conv_lstm | random | 0.8404 | 0.8610 | -0.16 | -0.87 |
| conv_lstm | south | 0.8635 | 0.8705 | -2.91 | -2.00 |
| lstm | distance | 0.8474 | 0.8625 | -1.05 | -1.24 |
| lstm | east | 0.8693 | 0.8852 | -3.66 | -3.89 |
| lstm | mountain | 0.8410 | 0.8601 | -0.28 | -0.95 |
| lstm | north | 0.8226 | 0.8446 | 1.92 | 0.86 |
| lstm | west | 0.8463 | 0.8628 | -0.91 | -1.27 |
| lstm | plain | 0.8394 | 0.8553 | -0.09 | -0.39 |
| lstm | random | 0.8385 | 0.8627 | 0.02 | -1.26 |
| lstm | south | 0.8616 | 0.8742 | -2.74 | -2.60 |

The models are designed to perform multi-step forecasting of wind speed (FF) over a 24-hour horizon. Specifically, each model produces a sequence of 24 predictions corresponding to hourly lead times from $h+1$ to $h+24$. This direct forecasting strategy enables the models to generate the entire prediction horizon in a single forward pass. Performance metrics are computed globally over the 24 forecast steps.

## 4 RESULTS

Table 2 summarizes the performance obtained for each architecture–strategy pair. For each combination, the table reports the mean RMSE and the mean gain defined in equation 6 with respect to the non-transfer baseline, averaged over horizons from $h+1$ to $h+24$ and computed under two historical windows (2015–2020 vs 2020 only).

Several clear tendencies emerge. First, the majority of strategies lead to either negligible or negative gains, with some configurations showing systematic performance degradation. This effect is particularly visible for directional dominant wind speed groupings such as East and South, which consistently result in average losses exceeding -2.00% across all architectures. Random selection, distance-based grouping and west also perform poorly, with mean relative gains close to zero or negative, reflecting unstable behavior.

In contrast, the North strategy stands out as the only configuration that provides positive average gains across all tested architectures. While these improvements remain modest (generally between +0.86% and +2.25%). This suggests that under specific transfer conditions, it is possible to mitigate

the negative effects observed elsewhere. Strategies based on topographic classification (mountain and plain) show intermediate results: they do not yield clear benefits, but their performance degradation remains less severe than for other directional groupings.

Overall, the table highlights the absence of a universally effective transfer strategy. Gains remain limited in magnitude and often offset by frequent negative transfers. Increasing the dataset size leads to better performance, although the improvements remain negligible in most cases. These findings confirm that, in the present setting, transfer learning cannot be considered a reliable approach for systematically improving model performance.

## 5 DISCUSSION

Our results demonstrate that, in the context of wind speed forecasting in Corsica, inductive parameter-based transfer learning rarely improves performance and frequently induces negative transfer. Contrary to a substantial body of literature reporting strong gains from transfer learning Oh et al. (2022); Zhu et al. (2025); Zhang et al. (2024), none of the geographically informed strategies evaluated here provide systematic improvements over training a model directly on the target station. Only the North-based grouping yields modest but consistent gains, suggesting that beneficial transfer may occur only under highly specific alignments between local atmospheric regimes.

A primary factor behind these outcomes is the pronounced distributional heterogeneity between stations. The KS-based heatmap in Figure 4 highlights substantial discrepancies in wind speed distributions across the network. However, the KS statistic captures only one-dimensional discrepancy. To obtain a more informative characterization of inter-station divergence, we extended this analysis by computing the Wasserstein distance between source and target distributions using Sinkhorn regularization. We then assessed the relationship between this multidimensional divergence and transfer performance through partial correlation, controlling for the no-transfer baseline as a covariate. This analysis yields a partial correlation of approximately $18\%$, indicating that large distributional differences contribute to negative transfer but do not fully explain it. Distributional shift therefore appears to be a necessary but insufficient predictor of transfer success.

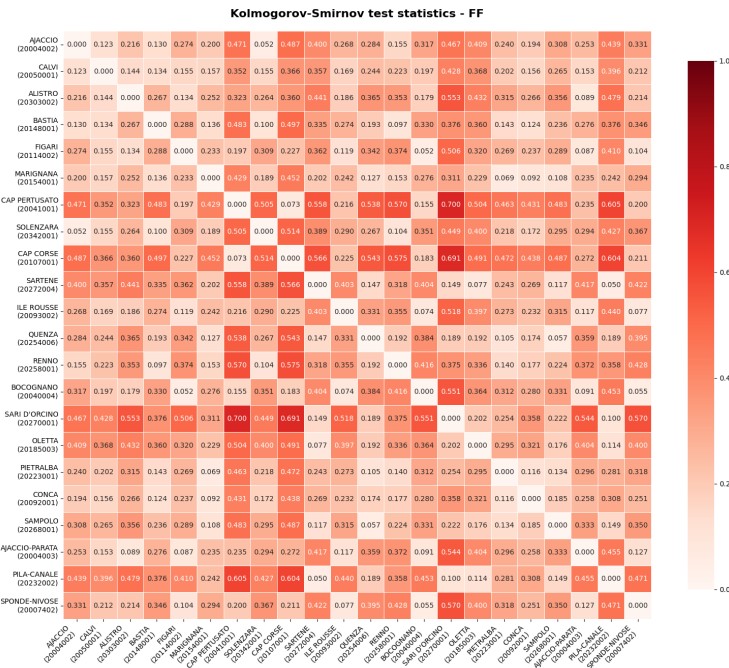

Figure 4: Heatmap of Kolmogorov–Smirnov test statistics highlighting inter-station distributional shifts in wind speed.

Beyond global distributional differences, several stations exhibit highly idiosyncratic temporal patterns. Models trained on such stations may overfit these localized structures and subsequently struggle to "unlearn" them during fine-tuning, even when distributional distances are not extreme. This provides a plausible mechanism for persistent negative transfer. Identifying such harmful patterns constitutes a promising direction for future research.

We also examined model performance separately across the 24 forecasting horizons. As shown in Figure 5, the RMSE increases predictably with the forecast horizon, but all architectures display almost identical profiles regardless of the input window size. This overlap indicates that extending the historical context (from 3h to 48h) does not yield systematic performance gains. More importantly, the horizon-wise evolution is nearly unchanged across models, suggesting that the effects of negative transfer are not concentrated at specific lead times but instead affect the entire prediction range uniformly.

The persistence baseline remains clearly inferior across all horizons, confirming that the deep models do extract meaningful short-term temporal structure, but this advantage does not translate into improved transferability between stations. A more granular, hour-specific analysis will help identify which forecasting horizons are most sensitive to transfer.

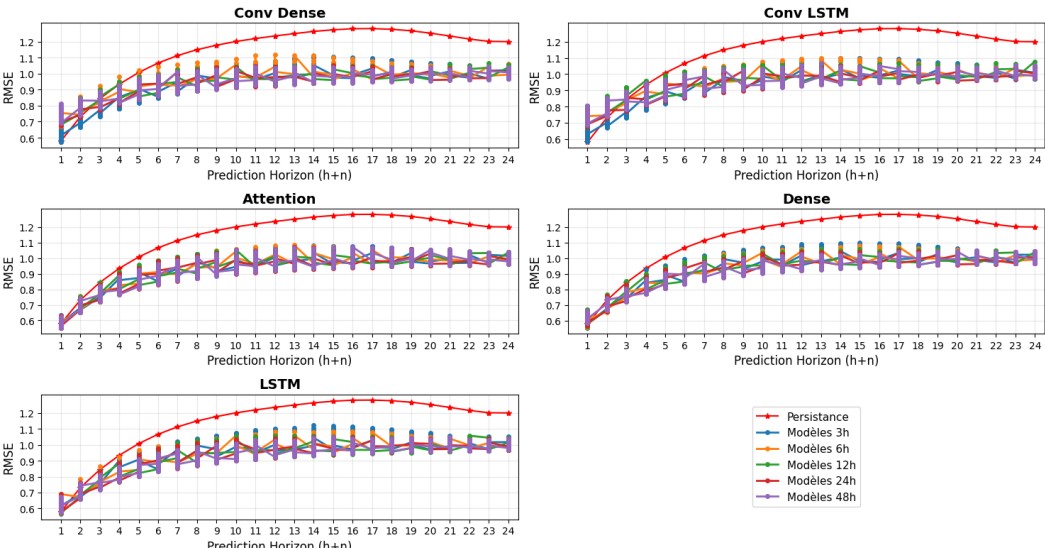

Figure 5: Horizon-wise RMSE across model architectures and input window sizes. The figure compares RMSE evolution over 24 forecasting horizons for five architectures and multiple input window sizes. Model performances largely overlap, indicating no systematic benefit from extending the historical context. Errors increase with horizon, while the persistence baseline remains consistently inferior.

Taken together, these findings call for cautious deployment of transfer learning in highly heterogeneous meteorological environments. They also highlight the need for distribution-aware, context-specific transfer criteria and for models capable of adapting to localized temporal and spatial structures.

## 6 CONCLUSION

This study evaluated multiple transfer learning strategies for wind speed forecasting in Corsica. Contrary to much of the literature, our results show that transfer learning does not systematically improve performance and often induces negative transfer. The main reason lies in the strong distributional wind speed heterogeneity between stations, driven by complex topography, as confirmed by KS tests. This not respect the assumption of source–target similarity Zhang et al. (2021), explaining the recurrent failures of transfer.

These findings call for caution: transfer learning cannot be assumed universally beneficial, especially in geographically heterogeneous contexts. Additionally, transferred models may converge toward suboptimal solutions that do not align well with the specific dynamics of the target station. Future work should prioritize distribution-aware strategies, hybrid approaches, h+6 forecasting, and spatially explicit models such as graph neural networks, while also reporting negative results to avoid overly optimistic conclusions. Meta-learning techniques should also be explored to improve model adaptability.

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
