# OpenReview forum: "Evaluating Inductive Parameter-Based Transfer Learning with Deep Neural Networks for Wind Forecasting in Corsica"
_ICLR.cc/2026/Conference — Submitted to ICLR 2026_

### Official Review · Reviewer_WcdZ · 2025-10-31

**Soundness:** 2
**Presentation:** 3
**Contribution:** 2
**Rating:** 2
**Confidence:** 4

**Summary:**

This study investigates the utility of several transfer learning strategies reported in past studies for wind speed forecasting across meteorological stations in Corsica. The experiments cover a wide range of architectures from recurrent models to attention-based models. The results show that transfer learning methods do not perform as well as expected and as reported in past studies.

**Strengths:**

- This paper investigates applications of deep neural network models to a meteorological prediction task.
- A detailed case study using Corsica wind speed was conducted. Various models are tested with several strategies.
- The experimental results suggest the limited gain from transformer learning, which counteracts the growing expectation (and reports) in this direction.

**Weaknesses:**

The analysis of this work appears solid but there are several critical concerns.

---
**Generality of the results**

This paper focuses on only one case study. While the experiments seem solid, it is hard to claim that transfer learning does not work well in meteorological experiments in general. Transfer learning may not perform well in the Corsica case, but it worked well in other cases in past work. The Discussion section explains this based on the complex topography of Corsica, but more general claims should be verified. Finding one "counter example", though important, is not impactful enough to meet the standards of ICLR in my personal view.

---
**Technical Contributions**

In my understanding, this study does not offer any technical contributions, not proposing a new model, evaluation framework, or training recipe to make transfer learning work in the Corsica case. The major contribution seems to be meteorologically meaningful, but the impact on the machine learning community is limited (even as an application paper).

---
I feel the paper is still half-way. This paper finds a challenge to address but does not propose any solution to it. The amount of experiments are limited (no appendix, even the main text is not completed).

**Questions:**

Please address the weaknesses.

---

> ### Author Response · Authors · 2025-12-03
>
> Thank you for your feedback and questions. We attempted to further develop the paper by adding a more in-depth analysis. We have computed a correlation coefficient between distributional distances and negative transfer. To achieve this, we used the Wasserstein distance with Sinkhorn regularization in order to account for the full characteristics of the distributions. Given the positive correlation, the greater the overall distributional distance, the more negative the transfer performance tends to be. However, this relationship does not explain all observed effects. We believe that further analysis of temporal patterns would be a promising direction to better understand the underlying causes.

---

### Official Review · Reviewer_deyX · 2025-11-01

**Soundness:** 2
**Presentation:** 2
**Contribution:** 2
**Rating:** 2
**Confidence:** 4

**Summary:**

The paper evaluates inductive, parameter-based transfer learning for wind-speed forecasting across 22 Corsican stations with long records ($n \approx 80000$). The authors train several neural architectures (FFN, LSTM, Conv-LSTM, Conv-Dense, encoder-only attention) and test station-to-station transfer based on four heuristics: distance, topographic class, dominant wind direction, and random. Performance is measured by RMSE over a 24-hour horizon (direct multi-step), with gains computed versus the same architecture trained without transfer. Across nearly all of the settings, structured transfer rarely helps and often hurts performance. The authors attribute this failure of transfer learning to strong cross-station distribution shift, uncovered via a KS-statistic heatmap. This is well supported by theoretical results on the viability of transfer learning.

**Strengths:**

1. Clear, negative-result study on a practical forecasting task. Systematic TL setups across multiple architectures, with transparent training/validation/test splits and a year-ahead test set (2023). Results call into question commonly held beliefs about transfer learning and its viability for modeling complex environmental processes.

2. Multiple, interpretable transfer heuristics, based on geography and wind regimes, help show that even very similarly located stations with similar wind patterns may not be transferable. The "random transfer" setting also serves as an interesting secondary baseline.

3. The Kolmogorov-Smirnov heatmap provides a plausible explanation for negative transfer and makes an explicit connection between domain shift and TL failure.

4. The wide range of architectures and transfer methods demonstrate that this phenomenon is not due to a single, poorly chosen or poorly trained model.

**Weaknesses:**

1. Limited insight into “why”. The analysis (Table 1 and discussion) largely stops at “large distribution shift implies no transfer.” While this finding is true on the whole, it does not help to quantify when and where transfer learning might still be helpful. This analysis would strongly benefit from dissimilarity-benefit plots e.g. how transfer gains vary with KS distance or forecasting lead time. Other possibilities include: frequency-domain comparisons (are low- or high-frequency components mis-transferred?) or seasonal/diurnal stratification (transfer may fail differently by season or hour of day). These types of analyses could help reveal the full scope of transfer learning's feasibility on this problem.

2. Limited scope. The study is confined to Corsica, whose wind patterns are strongly shaped by complex topography / orography and by highly regional flows (Mistral/Tramontane), which might limit the scope of these findings to Corsica. Whether these results would still hold in more settings with flatter terrain and consistent wind patterns, or even other mountainous regions, is not clear. This would need to be shown in a cross-region evaluation (e.g., mainland France), which could also help isolate whether transfer learning, as conduced here, works in relatively "easy" settings.

3. External validity. This work only considers transfer learning within Corsica, rather than externally from other regions. It would be quite interesting, and strengthen the claims of the article, if it were shown that a strong, previously published wind model (e.g. Transformer or GNN based approach (e.g. Mehrkanoon, (2024)) could not be transfer learned onto Corsica data. I.e. if the fine-tuned TL variant was no better than the Corisca only variant. Many of these approaches are spatiotemporal models, which would also help broaden the scope of the analysis.

4. Limited time lag. By considering only a very short time lag / context window (3 time steps), there is limited information for the neural networks to exploit. It may be possible that the models simply don't have enough information for TL to be effective. A longer history could allow for the model to capture diurnal/weekly cycles and low-frequency variability, which might be more amenable to transfer learning.

5. All experiments use per-station models, yet station series are spatially correlated. Spatio-temporal approaches are very commonly used for wind forecasting; including one such baseline would help determine whether negative transfer stems from parameter reuse or from the lack of spatial inductive bias. This could likely be included along with an external validation by transfer learning a pre-existing model.

**Questions:**

1. Given the KS plot in the discussion, is transfer learning in this situation in this manner possible? These results seem to suggest that no method may be able to overcome the high divergence between the neighboring series. I would be really curious to see if a spatiotemporal model can help here though.

2. Please add a single figure showing transfer gain vs a measured divergence (e.g., KS) for every source–target pair, plus a simple fit of the model to confirm they are fully trained. Also show the same plot per a few different lead times to see if this changes over the forecasting horizon (1…24h).

3. Can you fine-tune a strong published wind model, trained elsewhere, onto Corsica and compare to its Corsica-only version.

4. Add one spatio-temporal graph model (train from scratch and with your TL framework) to separate “parameter reuse is unhelpful” from “the model lacks spatial inductive bias.”

5. Repeat your main table with longer input histories (e.g., 12, 24, 72 steps) to test whether TL benefits change when the model can exploit diurnal/low-frequency structure.

6. Alongside the aggregated RMSE, can you break down these results into per lead time statistics? I'm wondering if some the TL benefits may only be seen over very short forecasting horizons.

---

> ### Author Response · Authors · 2025-12-03
>
> Thank you for your feedback and questions. We have addressed as many of your concerns as possible. Due to time constraints, we were only able to respond to the questions related to input steps and forecasting horizons (3h, 6h, 12h, 24h, 48h). We also computed a correlation coefficient between the global distributional distance (Wasserstein with Sinkhorn regularization) and the target RMSE. We plan to continue this work by investigating correlation patterns at a finer level of granularity.

---

### Official Review · Reviewer_DQsT · 2025-11-02

**Soundness:** 2
**Presentation:** 2
**Contribution:** 2
**Rating:** 4
**Confidence:** 4

**Summary:**

This paper evaluates transfer learning strategies for wind speed forecasting across 22 meteorological stations in Corsica using deep neural networks. The authors compare four transfer strategies (distance-based, topographic classification, dominant wind direction, and random) against non-transfer baselines using various architectures (LSTM, Conv-LSTM, attention-based, dense networks). The results show that transfer learning does not systematically improve performance and often leads to negative transfer. The authors attribute this to significant distributional differences in wind speed across stations due to Corsica's heterogeneous topography, confirmed by Kolmogorov-Smirnov tests. While the paper presents a negative result, it provides valuable insights into the limitations of transfer learning in geographically heterogeneous contexts.

**Strengths:**

1. The paper challenges common assumptions about transfer learning benefits, which is valuable for preventing over-optimistic deployment. The authors are transparent about limitations and don't oversell their results.
2. Testing multiple transfer strategies (distance, topography, wind direction, random) provides comprehensive coverage of intuitive approaches. Testing across 5 different neural network architectures strengthens the generality of findings within this context.
3. The KS test analysis (Figure 4) effectively demonstrates the distributional differences that explain negative transfer.

**Weaknesses:**

* The conclusions about transfer learning failure are presented as challenging, yet the evidence is drawn from a single geographic region. The paper does not demonstrate whether these findings apply to, tther island regions with similar topography, continental regions with comparable geographic heterogeneity, other meteorological variables beyond wind speed, or different forecasting tasks or temporal resolutions
* While the paper identifies distributional differences via KS tests as the primary cause of negative transfer, but doesn't establish a predictive relationship. Do larger KS statistics correlate with worse transfer performance? At what threshold does distributional similarity enable successful transfer?
* Beyond stating "distributional shifts," what specific aspects drive failure? Is it mean wind speed differences? Variance? Temporal patterns? Extreme value distributions? Understanding this would inform which similarity metrics actually matter.
* The paper reads as stopping at problem identification without attempting solutions. This significantly limits practical impact—practitioners learning that basic transfer fails in heterogeneous contexts need guidance on what does work.

**Questions:**

See above

---

> ### Author Response · Authors · 2025-12-03
>
> Thank you for your feedback and questions. We have computed a correlation coefficient between distributional distances and negative transfer. To achieve this, we used the Wasserstein distance with Sinkhorn regularization in order to account for the full characteristics of the distributions. Given the positive correlation, the greater the overall distributional distance, the more negative the transfer performance tends to be. However, this relationship does not explain all observed effects. We believe that further analysis of temporal patterns would be a promising direction to better understand the underlying causes.

---

### Meta-Review · Area_Chair_8dSF · 2026-01-07

**Summary:**

The paper highlights the challenges of transfer learning (TL) for wind speed forecasting and demonstrate that TL shows minimal improvement due to significant distributional differences. All reviewers agree that the paper points out a very important problem in TL for meteorological applications.

However, the concerns were uniform in that the paper makes little attempt to understand why, focuses on a specific test-case without comprehensively testing other scenarios, limited analysis on the possible failure modes, and concerns on the NN having insufficient information to be trained well. This limits the contribution of the paper and it currently reads as a partial analysis. There was no discussion from the authors and hence I recommend rejection.

**Reviewer Concerns:**

No rebuttal

**Reviewer Scores:**

Scores stand at 4,2,2 without rebuttal.

---

### Decision · Program_Chairs · 2026-01-26

Reject